# Particle size distribution: An experimental study using southern African reduction methods and raw materials

**Paloma de la Peña**[1,2,3]*, **Marc Thomas**[4], **Tumelo R. Molefyane**[5]

**1** McDonald Institute for Archaeological Research, University of Cambridge, Cambridge, United Kingdom, **2** Departamento de Prehistoria y Arqueología, Universidad de Granada, Granada, Spain, **3** Evolutionary Studies Institute, University of the Witwatersrand, Johannesburg, South Africa, **4** UMR 5608 TRACES, Université Toulouse Jean Jaurès, Toulouse, France, **5** Guías de Espeleología y Montaña (GEM), Madrid, Spain

* paloma.delapenya@gmail.com

## Abstract

We experimentally created a particle size dataset that is based on reduction sequences and raw materials typical of the Middle and Later Stone Age in southern Africa. The reason for creating this new dataset is that current particle size frameworks are based, almost exclusively, on flint and western European knapping methods. We produced the dataset using knapping methods and raw materials frequently encountered in the southern African archaeological record because we wanted to test whether it has the same distribution as particle size datasets experimentally created in Europe, and to initialise the production of a database for use in the analysis of lithic assemblages from southern African Late Pleistocene deposits. We reduced 117 cores of quartz, quartzite, jasper, chalcedony, hornfels, and rhyolite. The knapping methods selected were unidirectional, discoidal, *Levallois* recurrent and bipolar flaking. In this article we compare this new particle size distribution dataset with the results obtained from previous experiments. We found that the southern African dataset shows a wider size range distribution, which seems to be explained by differences in knapping methods and raw materials. Our results show that there is overlap between the distribution of the southern African experimental knapping dataset and the sorting experiment conducted by Lenoble on flint artefacts in a runoff context. This article shows that a particle size analysis is not sufficient on its own to assess the perturbation of an archaeological assemblage and must be coupled with other analytical tools.

## Introduction

Stone Age sites are complex sedimentary accumulations, the formation of which depends on a combination of anthropogenic and natural processes [1–4]. Natural processes, including sedimentary processes, have often altered the spatial organization, composition, and chronological signal of the archaeological record [5]. To measure this loss of information on sites, disturbance assessment studies need to be carried out. Any archaeological remains can undergo the

**Funding:** Regarding the funding, I (Paloma de la Peña) received operational support funding from the Center of Excellence in Paleosciences –National Research Foundation of South Africa for travel expenses. I also received a Pouroulis Family Foundation Fellowship in African Quaternary Archaeology and Hominin Palaeoecology at the University of Cambridge, which contributed to half of my salary during the research and writing process of this manuscript. The funders had no role in study design, data collection and analysis, decision to publish, or preparation of the manuscript.

**Competing interests:** The authors have declared that no competing interests exist.

effect of sedimentary processes, governed by physical laws [4, 6]. Thus, particle shape, size and density are the main physical criteria used to measure the degree of perturbation of the archaeological record [2]. The particle size analysis of lithic artefacts was first applied in Africa by K. Schick [2]. She pointed out that in many cases the sites are associated with sedimentary contexts affected by water (alluvial environments, lake margins, basins, marine beaches or even spring deposits) [2]. In her seminal work, K. Schick [2] conducted experiments to understand the spatial dispersion and size-class distribution (maximum length of artefacts considered) of lithic industries affected by water flow. She concluded that water flow can substantially, if not radically, affect archaeological assemblages [7, 8]. In addition, other natural processes are likely to have an impact on the particle size distribution of lithic assemblages, such as solifluction [9, 10], trampling [11–18], and aeolian processes [19].

Other studies have extended and followed a similar methodology to Schick's work, applying it mainly to European Palaeolithic contexts, particularly in the southwest of France [4, 20, 21]. A. Lenoble and P. Bertran developed a method based on sieving instead of manually measuring all lithic artefacts [4, 20, 21]. Based on the observation that rock knapping consistently caused the same particle size distribution, Lenoble and Bertran and colleagues proposed to compare an experimental knapping assemblage (from [21] called "EU database" for ease of reading in this article) to an experimental one subjected to overland water flow (from [4] called "RUNOFF dataset" for ease of reading in this article) [4, 20, 21] (see S1 File). The particle size distribution of these two sets of experiments, "EU dataset" and the "RUNOFF dataset" show separated distributions in a ternary plot (Fig 1). The knapping experiments were done mainly on flint and quartzite and included blade production, *Levallois* and discoid flaking, and bifacial shaping. The authors also added microlithic reduction sequences inspired by Palaeo-Eskimo knapping methods from the eastern Canadian Artic (called "ESKI dataset" for ease of reading in this article, from [22]).

In all knapping experiments, the distribution sizes follow a Weibull pattern, being long-tailed and positively skewed [23] with a proportion of largest pieces tending towards 0 [2, 4, 20–23], whereas the results of the "RUNOFF dataset" highlight different kinds of distributions according to where one stands in the runoff system. This clear distinction between both experiments in the final model allows one to test the size sorting of lithic archaeological assemblages ([20, 21], Fig 1). However, this experimental framework mainly includes debitage methods and raw materials typical of European contexts. Thus, the use of both datasets to study African assemblages is questionable, as different raw materials and knapping methods were used in Africa in the past.

## Questions and objective

In this paper, we contribute experimental research to the field of lithic technology and site formation analyses, focusing on southern African knapping methods and raw materials. The reason for creating this specific experimental dataset was to assess if the previous experimental datasets created in Europe for Middle and Upper Palaeolithic archaeological contexts (such as EU and RUNOFF datasets [4, 20, 21]), and mainly based on flint (and to a lesser extend quartzite) [4, 20, 21], were useful or not for the understanding of southern African archaeological lithic assemblages. We wanted to test if similar patterns of particle size distribution would be reproduced in southern African assemblages. The reason for this specific enquiry is that in southern African late Pleistocene contexts there are different methods of debitage and raw materials from those used in Europe. We hypothesized that different knapping methods and raw materials would create different proportions of lithic particles. For this purpose, we performed a large experiment on particle size analysis focusing on southern African rock varieties

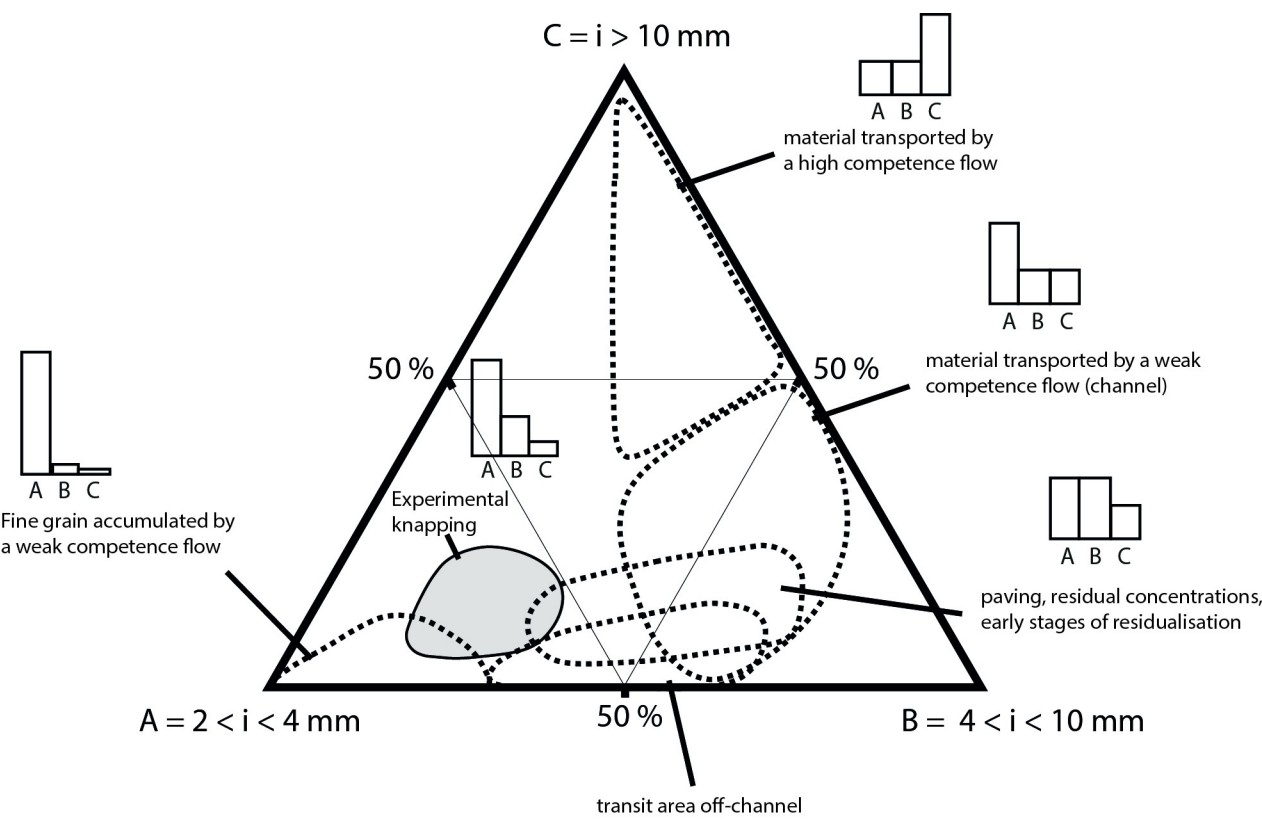

**Fig 1. Particle size evolution of a lithic assemblage affected by overland water flow from the experiment of Lenoble [4].** The light brown area is the distribution of the EU dataset [20, 21]. The dashed areas correspond to the size class distribution in the RUNOFF experiment [4]. The "A" triangle summit corresponds to the [2; 4] mm width size class percentage, the "B" triangle summit to the [4; 10] mm width size class percentage and "C" to the [10;+∞ mm width size class percentage]. From [4].

(automorphic and xenomorphic quartz, quartzite, jasper, chalcedony, hornfels and rhyolite) and knapping methods typical of the Middle and Later Stone Age records (unidirectional, bipolar knapping, discoid, recurrent *Levallois*). The knapping methods selected are relatively common in Middle and Later Stone Age assemblages (*vid.* [24, 25] *inter alia*). The raw materials are the most common rocks used for knapping in the interior parts of southern Africa for the Late Pleistocene. The exceptions to this selection are dolerite and silcrete, which were excluded from this analysis because dolerite is an extremely difficult rock to knap and silcrete does not occur in any of the sites that we currently study.

This new experimental dataset (called "SA dataset" in this article, see S1 File) will be useful for our own research on Middle and Later Stone Age sites and offer an available dataset for other archaeologists working on African material. Additionally, this new investigation serves as a control experiment that is comparable to previous European publications on this type of research [20, 21]. Moreover, this new dataset will test if different raw materials and knapping methods, such as bipolar knapping and unidirectional reduction, generate different results from the previous experiments [20, 21]. The new results and the comparison with previous experiments will ultimately serve to study in depth and without bias the archaeological assemblages from southern Africa. Our immediate goal is to compare the results of this experimental research with archaeological lithic assemblages from sites such as Mwulu's Cave, Border Cave and Marshill (studied by one of the authors, PdlP). Our research questions are:

1. What is the particle size distribution specific to southern African raw materials and knapping methods?

2. Are there statistically significant differences regarding raw materials?

3. Are there statistically significant differences regarding the knapping methods?

4. Is the particle size distribution of the SA dataset different from the EU dataset?

5. Is the particle size distribution of the SA dataset different from the RUNOFF dataset?

The null hypotheses are that there is no statistical difference between the different raw material and knapping method datasets compared in this study, and compared with previous published datasets [4, 20, 21].

## Material and methods

One of the authors (PdlP) conducted all of the experimental knapping, having several years of stone tool making experience. The experiment includes the most common southern Africa rocks and lithic reductions used during the Middle and the Later Stone Age (*vid. supra)*. They are summarized in Table 1 with the number of cores reduced for each one of the knapping methods and raw material. In Fig 2, we show schematically all the knapping methods deployed in this analysis. In total 117 cores were knapped for this experiment (see S1 File). In Fig 3 the main raw materials are illustrated, alongside a selection of tools made during the experiments.

During the experiments, we tried to reduce the core until exhaustion, in other words, until no more flakes could be extracted. For the knapping technique in this experiment, we used a hard mineral percussion with a banded iron pebble from KwaZulu-Natal and a soft mineral hammer, a sandstone pebble from the Eastern Cape.

The specific knapping methods selected are as follows:

- Bipolar knapping: A method in which the core is placed on an anvil and held with the bare hand. The rock is struck from above with a hammer held in the other hand, causing blanks to fly off from the top and from the edge that is in direct contact with the anvil [29, 30]. Bipolar knapping has been reported in the Middle Stone Age and Later Stone Age [25].

- Discoidal Knapping: In this experiment we used what Terradas [31] named multifacial discoidal knapping (called "discoidal knapping"), which is equivalent to "centripetal knapping" according to V. Mourre [32]. In this sub-variant the knapper searches for 45 degree angles to obtain flakes until the core is exhausted, changing the striking platform almost every time and using the removals as such.

- Unidirectional knapping: In this method, we tried to exploit as much as possible from one striking platform following a unidirectional sequence of removals. Once the knapping surface from the first striking platform was exhausted, we rotated the core and continued knapping from a different striking platform, following the same strategy if possible.

**Table 1. Synthesis of the knapping methods and raw materials used in this experiment (A-quartz = Automorphic quartz, X-quartz = Xenomorphic quartz).**

|  | Chalcedony | A-quartz | Hornfels | Jasper | Quartzite | Rhyolite | X-quartz | Total |
|---|---|---|---|---|---|---|---|---|
| **Bipolar** | 8 | 4 |  | 10 |  |  | 10 | **32** |
| **Discoidal** |  |  | 10 |  | 10 | 10 | 12 | **42** |
| *Levallois* | 4 |  | 16 |  |  |  |  | **20** |
| **Unidirectional** | 10 |  | 8 | 5 |  |  |  | **23** |
| **Total** | **22** | **4** | **34** | **15** | **10** | **10** | **22** | **117** |

# FREEHAND

# AXIAL BIPOLAR

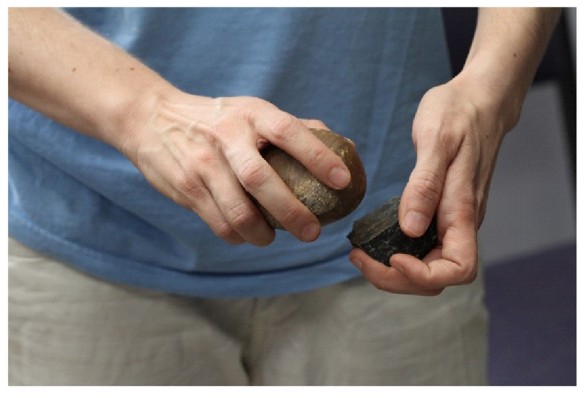

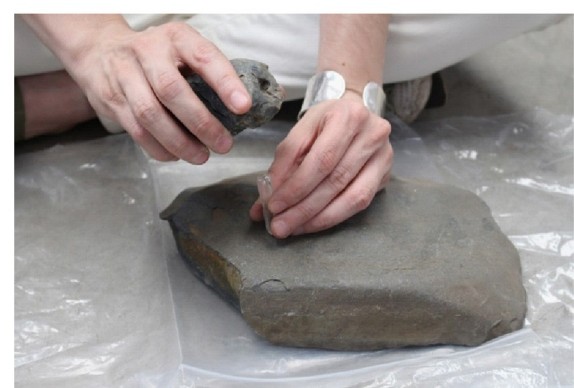

## Unipolar

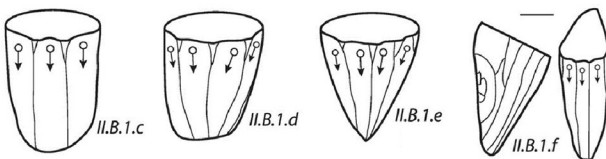

Modified from Shea, 2020

## Recurrent centripetal

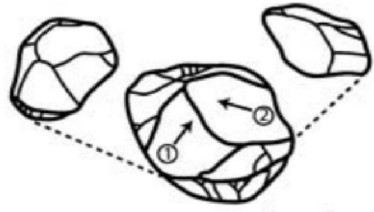

Modified from Tryon et al.,2005

Modified from de la Peña, 2015

## Multifacial discoidal

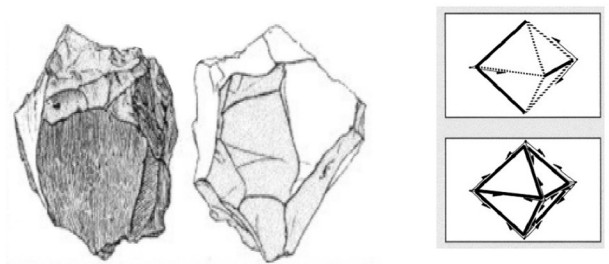

Modified from Terradas, 2005

**Fig 2. Examples of middle and later stone age reduction sequences employed in this knapping experiment: Discoidal, *Levallois* recurrent, unidirectional, and axial bipolar.** The figures are modified from [15, 26–28]. On the left freehand variants and on the right bipolar axial variant towards miniaturization. Photos: Paloma de la Peña.

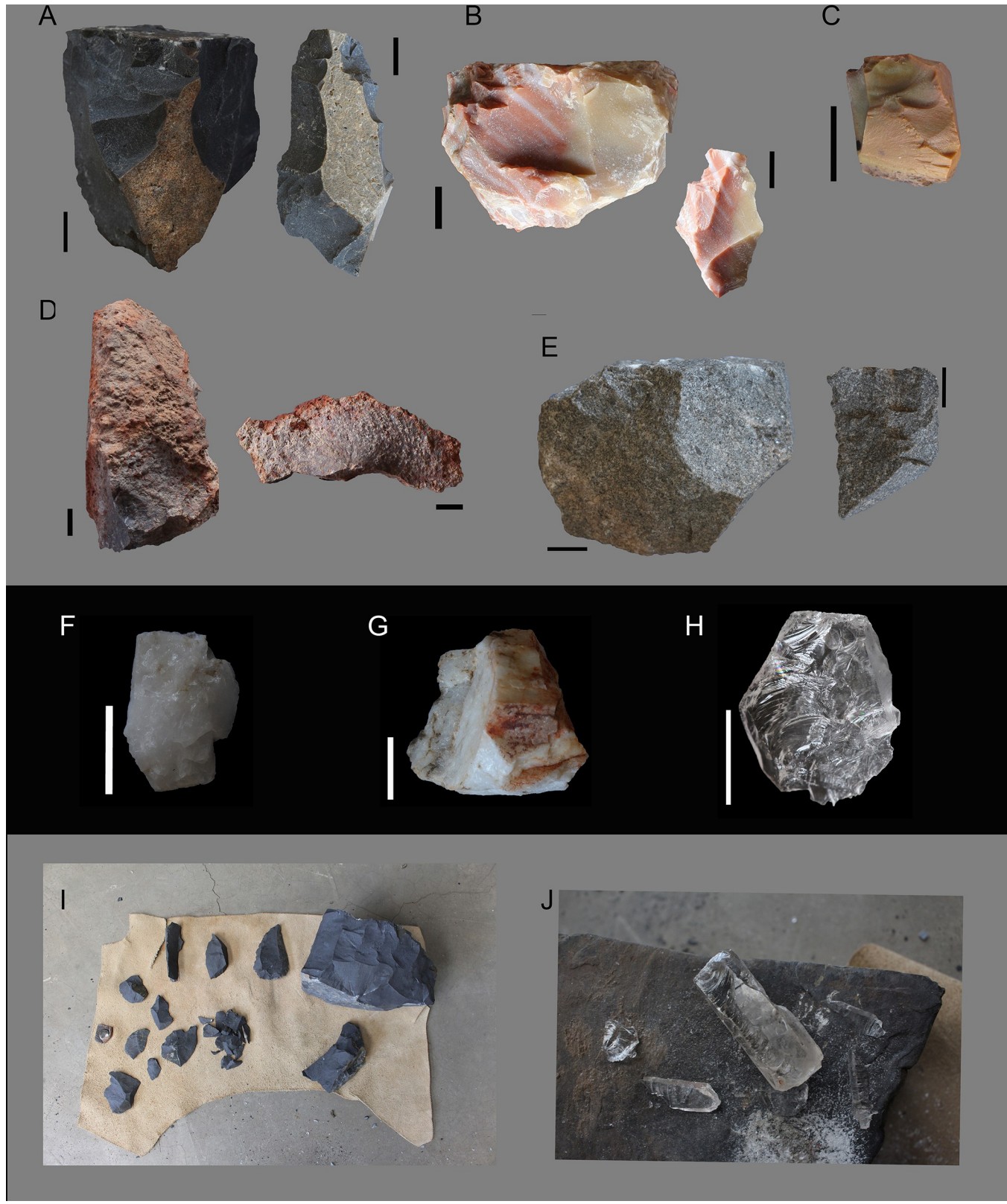

**Fig 3. Raw materials used for this experiment.** A. Hornfels (unidirectional core and flake). B. Jasper (unidirectional core and flake). C. Chalcedony (chunk). D. Rhyolite (core fragment) and flake (ventral view). E. Quartzite (discoidal core and flake, dorsal view). F. Xenomorphic quartz (bipolar core). G.

Xenomorphic quartz (core fragment). H. Automorphic quartz (bipolar core). I. Selection of blanks and cores in hornfels during one of the unidirectional reduction sequences. J. Core and blanks in the middle of the reduction through bipolar axial knapping of one of the automorphic quartz cores. All scales are 1 cm except for H, which is 5 mm. All the material presented is from this experiment, except for C and H, which were previously produced. Photos: Paloma de la Peña.

- *Levallois* recurrent (also referred to in the African literature as 'prepared cores'): In our experiments we followed the guidelines explained by E. Boëda [33]. Sometimes during reduction, we could extract two series of *Levallois* flakes. In some cases, we finished the knapping of these cores using discoidal reduction. This was done to obtain the most out of the volume of the rock in the final stage, and only when there was no possibility to prepare two hierarchical knapping surfaces in a *Levallois* fashion.

  The raw materials used for the experiments (Fig 3) include:

- Hornfels, a fine- to medium-grained metamorphic rock that is formed by contact metamorphism on the margins of igneous intrusions that crosscut the country rock [34]. The two varieties of hornfels used for this study come from two different dolerite dykes that intruded shales, one in the Eastern Cape and another in the Northern Cape. The variety used from the Eastern Cape was obtained from a river cutting of a massive hornfels contact metamorphism layer. Large chunks of the material were extracted and broken into smaller blocks for the experiments. The hornfels from the Northern Cape came in small slabs, usually of less than 10 cm for their longest axis.

- Chalcedony, a cryptocrystalline quartz [34]. The one we used is not banded, and in South Africa it mostly occurs in white, gray, black, or blue shades. Chalcedonies fill cavities and line fissures in rocks. In the Drakensberg and Lebombo Group, they occur in both basalts and rhyolites. Sometimes chalcedony contains a monoclinic polymorph called moganite [34]. The chalcedony of this experiment comes entirely from Siberia Farm, near a rock art site also known in the area as Franshoek. It is an outcrop with varieties of rocks ranging in colour from green to orange.

- Jasper, a chert coloured by the presence of iron [34]. The variety used for this experiment was banded red jasper from the Eastern Cape.

- Automorphic quartz (crystal quartz, called A-quartz in this article, following [35]), a euhedral mineral with the shape of a single large hexagonal crystal terminating alternately in major and minor rhombohedral faces [36]. Its flat faces are well developed around a symmetrical axis and bounded by rectilinear ridges [37, 38]. The crystal quartz used for this experiment comes from a small vein in Limpopo Province, from the surroundings of Mwulu's Cave [39], specifically from the Black Reef Formation.

- Xenomorphic quartz (following [35], called X-quartz in this article), a solid polycrystalline aggregate with an anhedral texture formed by a close-packed mass of crystals that are not usually bounded by their own crystal faces, but have their outlines pressed on them by the adjacent crystals [38, 40]. All the xenomorphic quartz used for this experiment comes from the Cradle of Humankind (Gauteng) area, near the site of Swartkrans.

- Quartzite, a metamorphosed sandstone. All the Quartzite used in this experiment comes from the Black Reef Formation in the surroundings of Mwulu's Cave [39] in Limpopo Province.

- Rhyolite, an extrusive igneous rock with a high silica content. Rhyolite is made up of quartz, plagioclase, and sanidine, with minor amounts of hornblende and biotite. Trapped gases

often produce vugs in the rock. These often contain crystals, opal, or glassy material. The rhyolite used for this experiment comes from the immediate surroundings of Border Cave in the Lebombo mountains (KwaZulu-Natal, South Africa).

To enable comparison of the southern African experimental knapping dataset with the RUNOFF dataset [4] and the EU dataset [20, 21] we have used the same size classes. The previous experiment used mesh of 2, 4, 5, 10, 20, 31.5 and 50 mm. After gathering all the particles produced during the knapping process, we hand-sieved the material in the same way recommended by Bertran et al. [20, 21]. As particle size analysis is time consuming, to speed up the counting for material-rich assemblages, the weights of small particles (2 and 4 mm) were obtained using mass/count ratios. We thus counted all the particles per size class and weighed them.

To describe our dataset and compare it with previous data we used ternary plots made with Past3 [41]. Additional statistics were made using the method developed by Weaver et al. [42] to obtain 95% confidence intervals in ternary plots (Fig 4). To compare the different datasets, we use Principal Component Analysis (PCA) performed under R using the FactoMineR package [26]. The MANOVA tests were run using Past3. The statistical descriptions of the PCA analysis are available in S2 and S3 Files.

Raw material and reduction strategies choices were made to follow as closely as possible the southern African archaeological context (*vid. supra*). All knapping methods are not applied to all raw materials, as for example there are certain knapping methods that never appear on certain raw materials in the southern African archaeological record. For instance, to the best of our knowledge, there is no *Levallois* reduction on automorphic or xenomorphic quartz in archaeological assemblages and, in the same vein, there is no record of bipolar knapping on anvil for hornfels. The number of experiments for each one of the variants explored per raw material is not consistent, as in some cases during the experiment the bad quality of the raw material (because of flaws and natural fractures) did not allow us to complete a reduction sequence. We tried to have at least five examples for each one of the raw materials and debitage variants and, ideally, to get 10 examples. This was not possible for all the variants selected (Table 1) because of the reasons explained. Thus, we assume statistical bias in the comparisons we performed as each raw material has not been knapped through each knapping method. Accordingly, it is not possible to measure the exact effect of these two last variables on particle size distribution. The aim being to compare archaeologically compatible options.

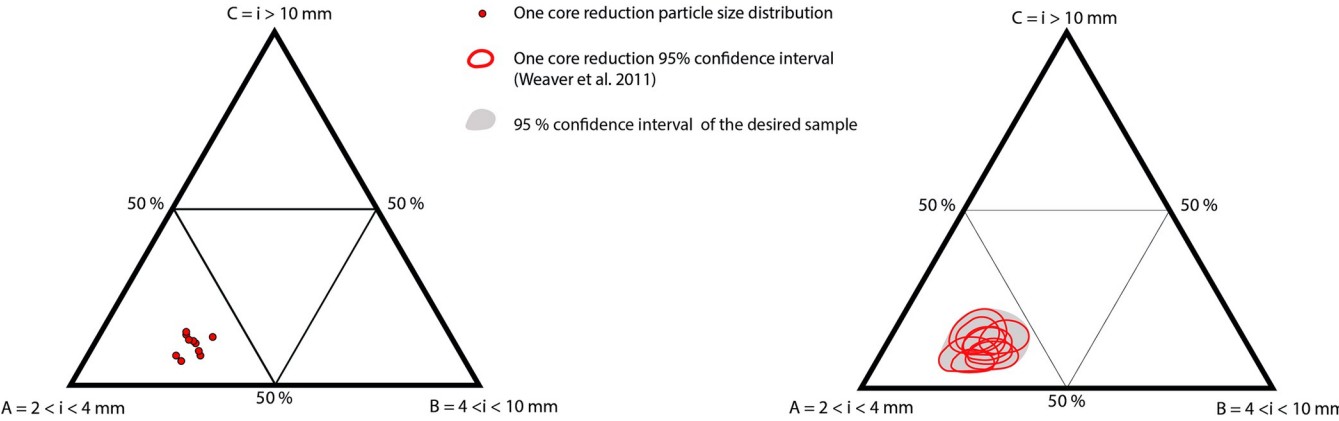

**Fig 4. Illustration of how we applied the Weaver et al. (2011) method to create 95% confidence ellipse intervals in the ternary plots.**

## Results

### What is the particle size distribution resulting from southern African raw materials and knapping methods?

In S1 File we present the particle size composition of the SA dataset, describing the size classes and their weights (the ESKI, RUNOFF and EU datasets are also included for comparisons). The histogram of the percentages per size class, considering knapping methods and raw materials, clearly shows that for all the raw materials and knapping methods there is a gradual decrease in particle numbers, as size classes increase. In all the series, the 5 mm particle class is slightly more abundant than the 4 mm size class (Fig 5). In all of the experiments, whether considering raw material or knapping method, the size classes 2 to 4 mm constitute around 40–60% of the particles (Fig 6).

The ternary diagrams (Fig 7) show a broad distribution of the SA dataset on experimental debitage size class distribution, predominantly composed of particles between 2 and 4 mm and between 4 and 10 mm. The 95% confidence interval calculated for each debitage group considerably increases the range of distribution for each raw material and knapping method.

In the ternary plots (Fig 8) rhyolite, quartzite and hornfels have a similar distribution towards the centre of the charts. Regarding knapping methods, unidirectional, discoidal, and *Levallois* seem to partially overlap, irrespective of the raw material considered (Fig 9). Furthermore, bipolar knapping produces the smallest fractions (between 2 and 4 mm) irrespective of the raw material. Freehand knapping (unidirectional, discoidal and *Levallois*) and bipolar knapping distribute slightly separately in the ternary plot.

### Are there statistically significant differences between the raw materials?

The factorial space of the "Raw material" principal component analysis (Fig 10 and S2 File) shows a clear opposition (mostly first axis, 39%) between, on one hand, chalcedony, jasper,

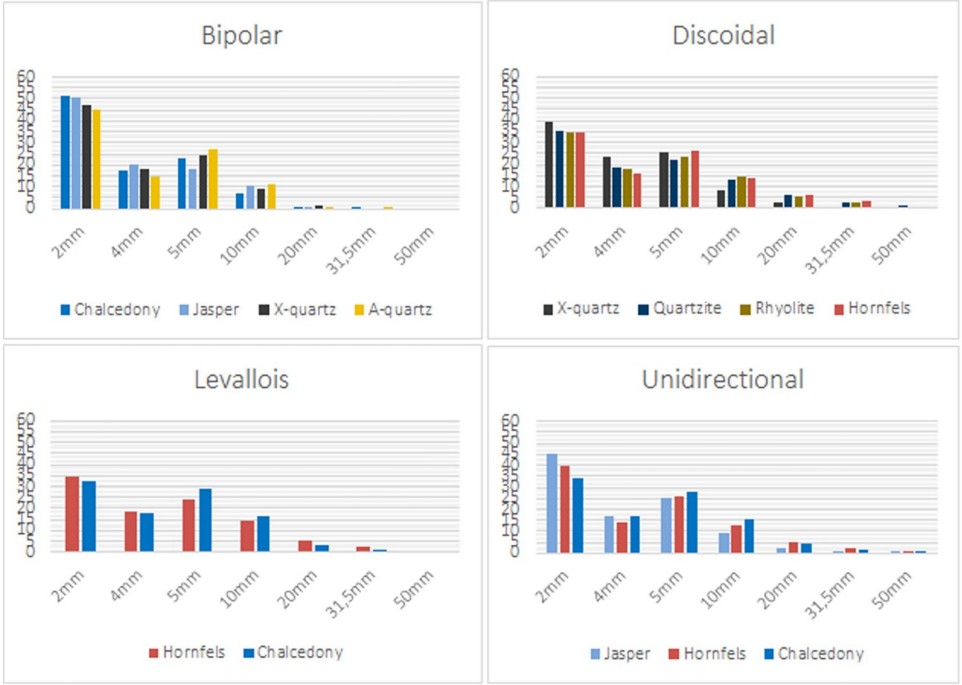

**Fig 5. Histograms comparing mean distributions of percentage of particles per size class (counts).**

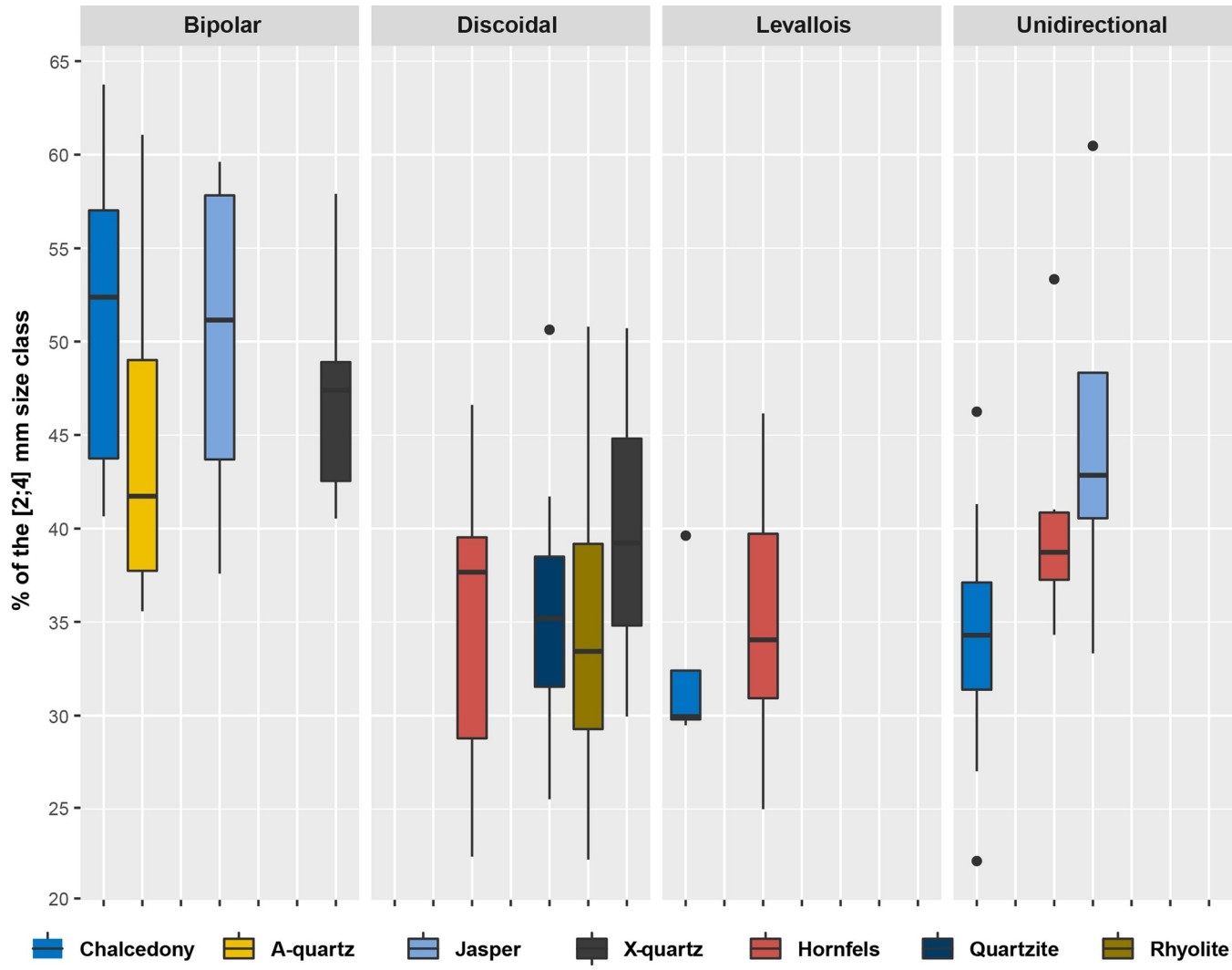

**Fig 6. Box plots comparing the proportion of the [2:4] mm size class (counts) depending on the raw material and the knapping method.**

X-quartz and A-quartz, and, on the other hand, hornfels, quartzite and rhyolite. The latter produced larger pieces, but have not been reduced through the bipolar technique. These differences are confirmed by a MANOVA test (using a pairwise comparison) with the raw material as an independent variable and the size distributions as dependent variables (Table 2). A group composed of chalcedony, jasper and x-quartz differs from hornfels, quartzite and rhyolite.

To ensure the reproducibility of all the statistical analyses of this research, the codes needed to perform the principal component analyses are provided in an rmarkdown script accessible in the S3 File.

## Are there statistically significant differences between the knapping methods?

The factorial space of the "Knapping method" principal component analysis (Fig 11 and S2 File) shows a clear opposition on the first dimension (39% of the variability) between methods producing smaller particles (2 to 4 and 4 to 5 mm width size class) and those producing larger

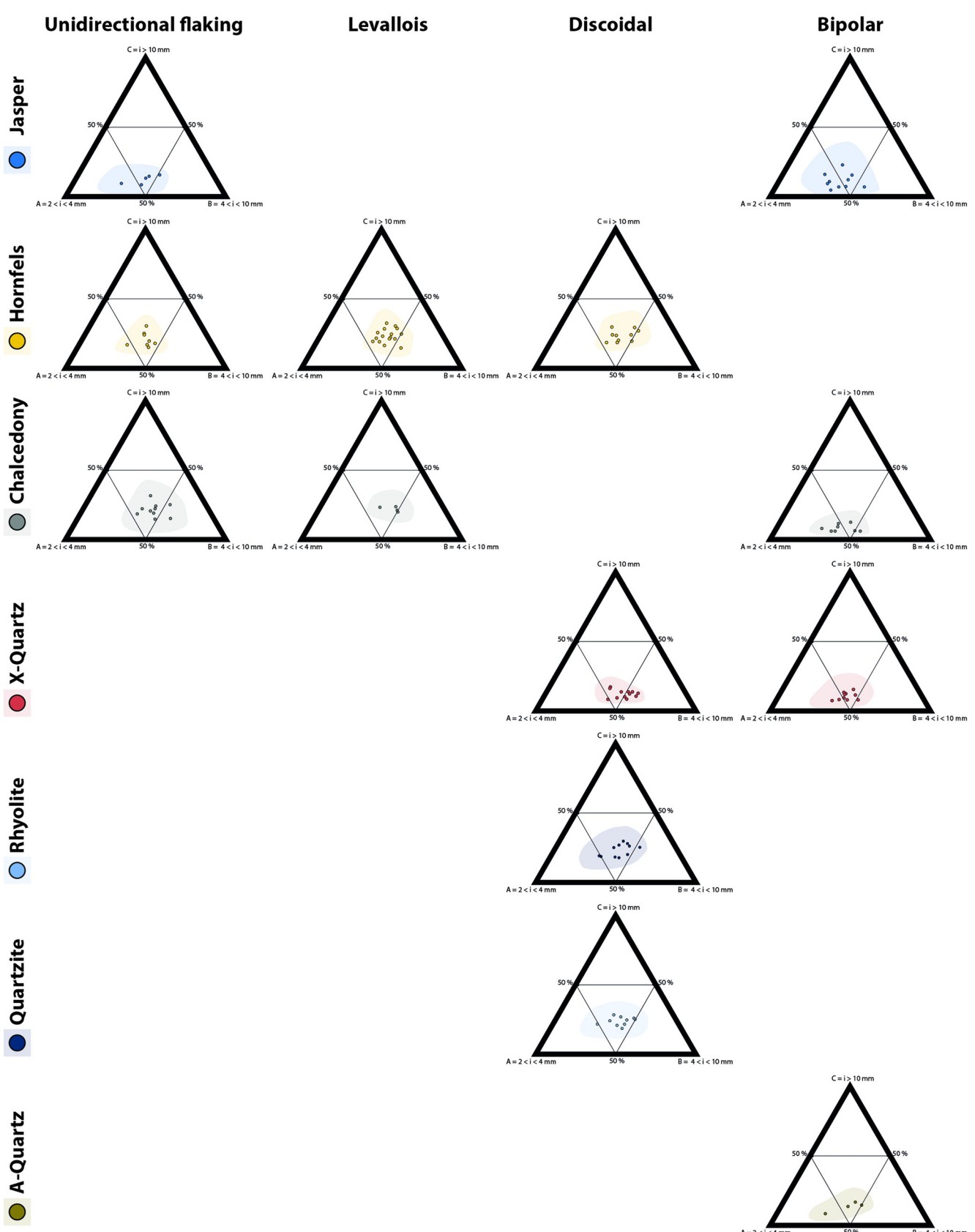

**Fig 7. Ternary plots per raw material and knapping method in the SA dataset (counts).** Shaded areas are 95% confidence intervals.

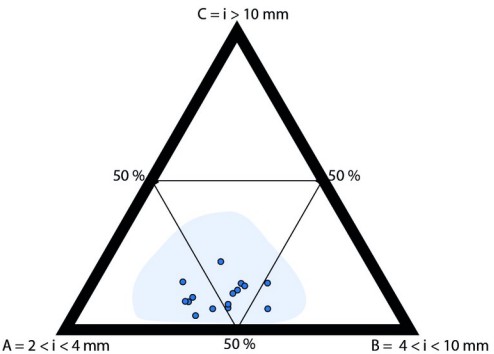

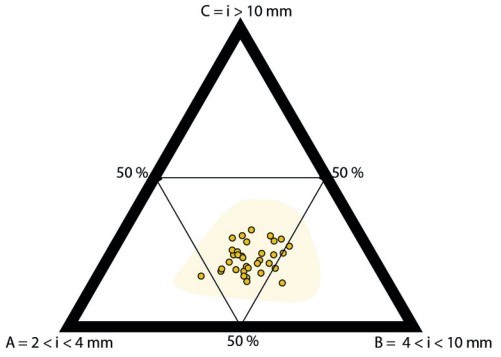

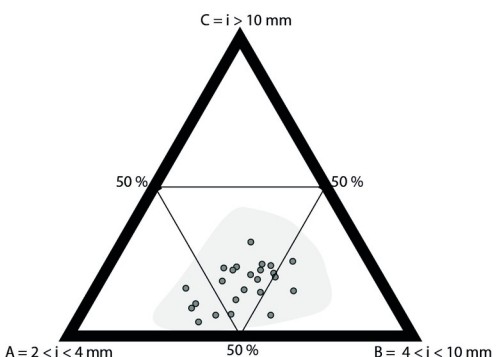

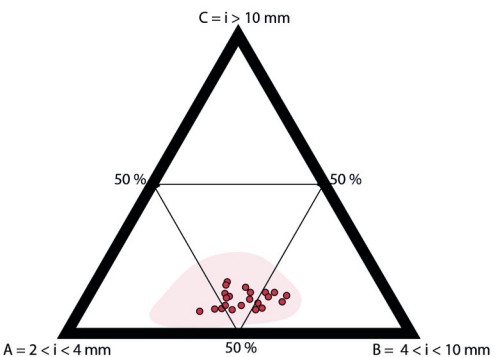

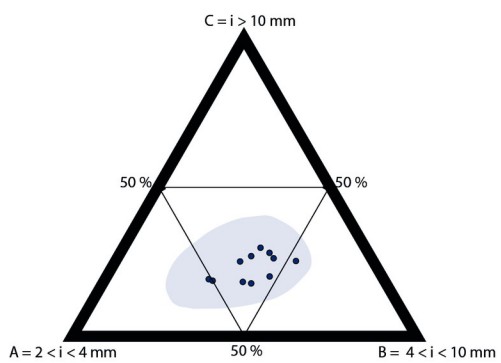

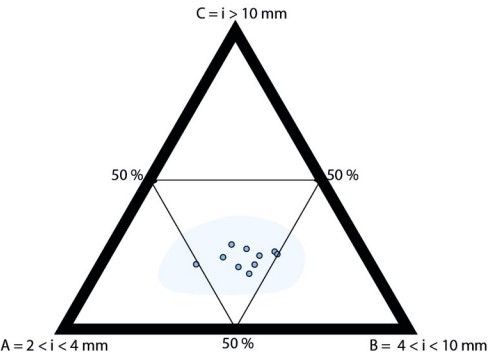

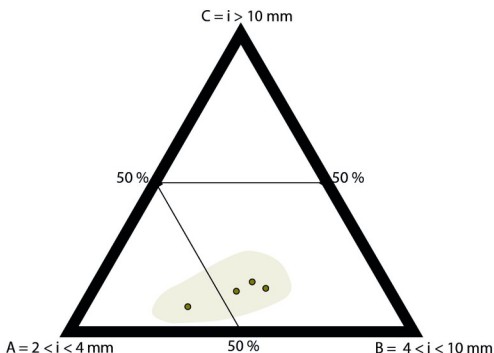

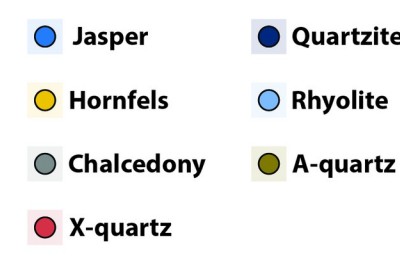

**Fig 8. Ternary plots of the particle size distribution per raw material (counts) in the SA dataset.** Shaded areas are 95% confidence intervals.

particles (over 10 mm wide). Bipolar knapping is best described by the size range variables 2 to 4 mm and 4 to 5 mm, as opposed to *Levallois*, unidirectional and discoidal methods, the distribution of which seem similar and produce larger pieces. These differences are confirmed by a MANOVA test with "Knapping method" as an independent variable (Table 3). Bipolar differs from all the other knapping methods, which have a size distribution that tends to be the same.

When combining knapping methods and raw material (Fig 12 and S2 File), three groups appear in the factorial space. The first one is best described by small particles (2 to 4 mm), and is composed of X-quartz discoidal knapping, all bipolar on anvil, and all jasper knapping experiments. The second group is best described by larger particles. It is composed of all hornfels, quartzite and rhyolite raw materials regardless of the associated knapping method. A third group, composed of chalcedony associated with *Levallois* and unidirectional flaking, is described as intermediate width particles (5 to 10 mm size class).

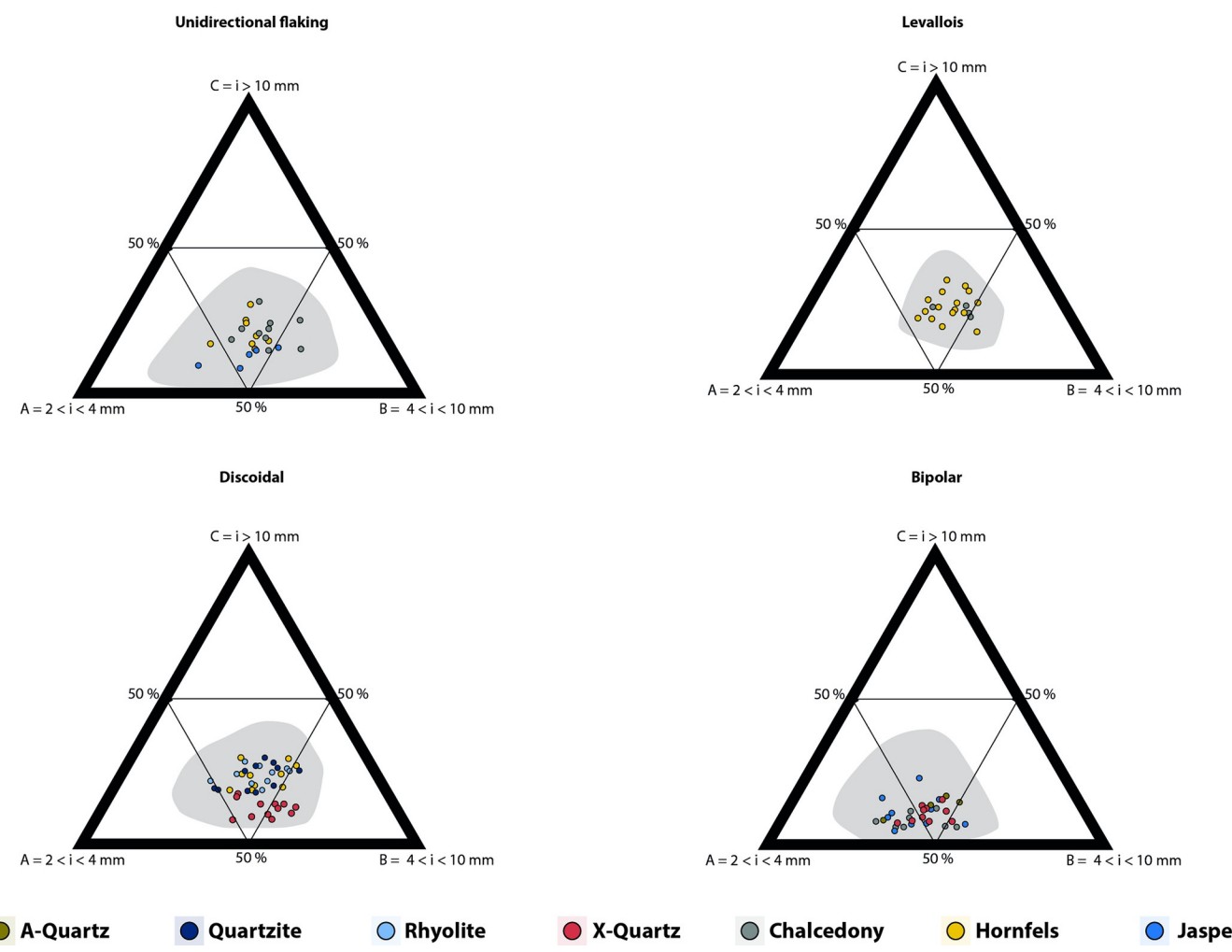

**Fig 9. Ternary plots of the particle size distribution per knapping method (counts) in the SA dataset.** Shaded areas are 95% confidence intervals.

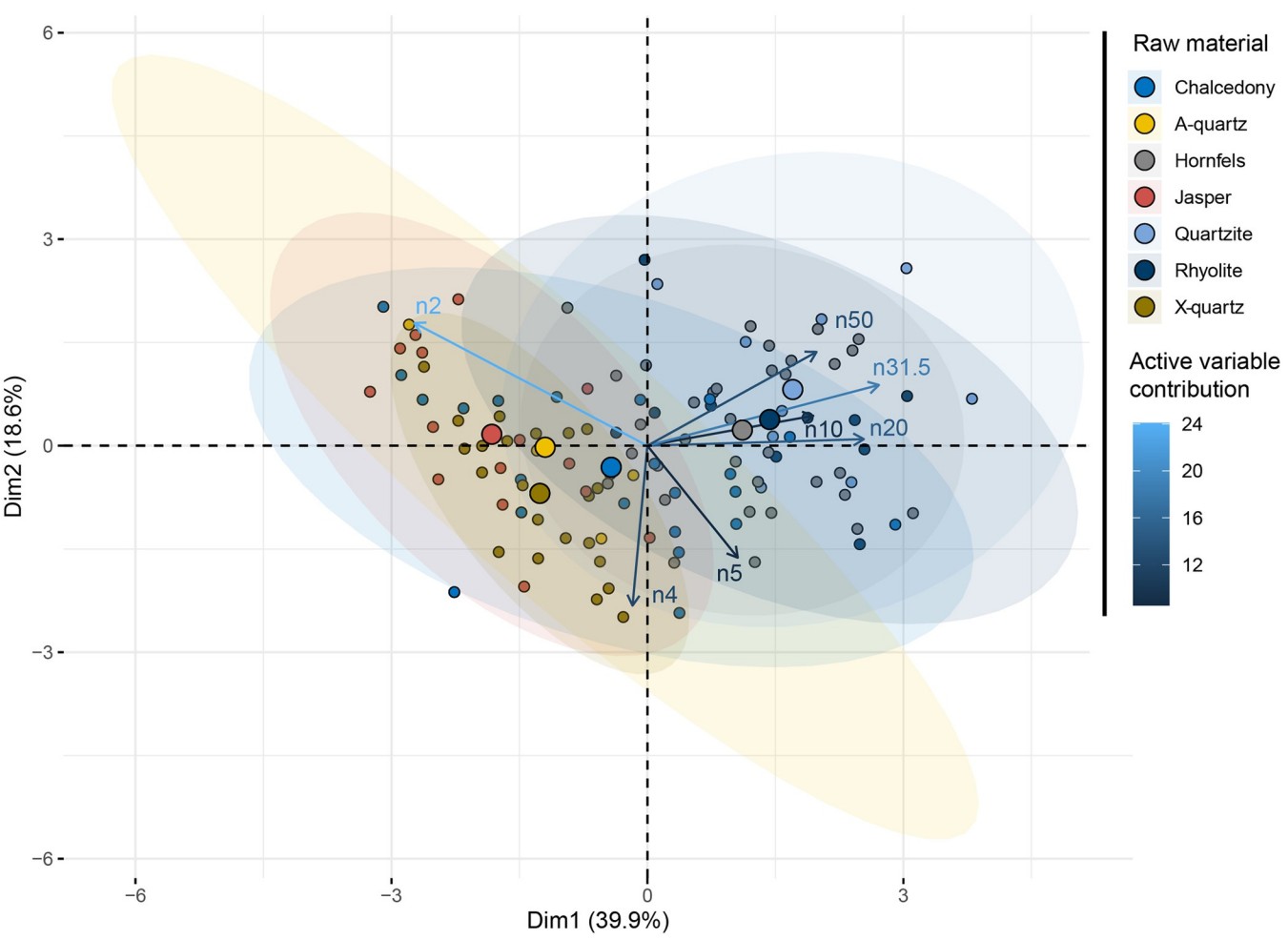

**Fig 10. PCA of the SA dataset with the raw material as a supplementary variable.**

## Is the particle size distribution of the SA dataset different from the EU dataset?

In the ternary diagram we see great differences between the three experimental knapping datasets on particle size distribution (Fig 13). This variability is confirmed by the principal component analysis including the EU dataset on particle size distribution [20, 21] and the SA dataset (Figs 14 and 15). When considering the knapping methods (Fig 14 and S2 File), three main

**Table 2. Results of the pairwise comparison from the MANOVA test.** Raw material is the independent variable, while size classes are the dependent variables.

|  | Chalcedony | A-Quartz | Jasper | X-quartz | Hornfels | Quartzite | Rhyolite |
|---|---|---|---|---|---|---|---|
| **Chalcedony** |  | 18,09 | 0,16 | 1,51 | *0,00* | *0,00* | *0,02* |
| **A-Quartz** | 18,09 |  | 11,61 | 12,17 | 0,14 | 1,22 | 3,50 |
| **Jasper** | 0,16 | 11,61 |  | 9,63 | *0,00* | *0,00* | *0,01* |
| **X-quartz** | 1,51 | 12,17 | 9,63 |  | *0,00* | *0,00* | *0,00* |
| **Hornfels** | *0,00* | 0,14 | *0,00* | *0,00* |  | *0,01* | 13,83 |
| **Quartzite** | *0,00* | 1,22 | *0,00* | *0,00* | *0,01* |  | 9,40 |
| **Rhyolite** | *0,02* | 3,50 | *0,01* | *0,00* | 13,83 | 9,40 |  |

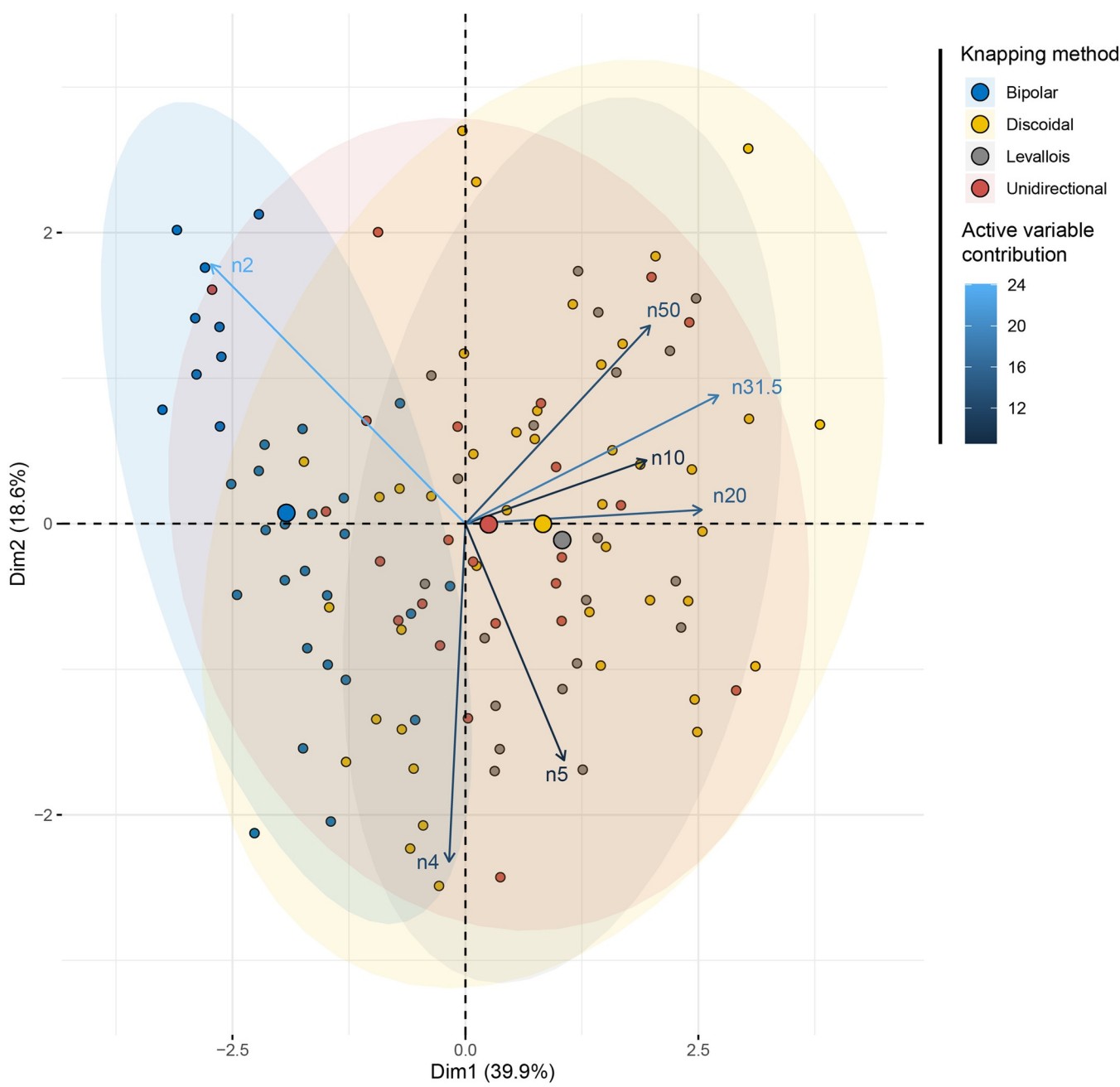

**Fig 11. PCA of the SA dataset with the knapping method as a supplementary variable.**

**Table 3. Results of the pairwise comparison from the MANOVA test.** Knapping method is the independent variable, while size classes are the dependent variables.

| | bipolar | discoidal | Levallois | unidirectional |
|---|---|---|---|---|
| **bipolar** | | *0,00* | *0,00* | *0,00* |
| **discoidal** | *0,00* | | 2,48 | 0,11 |
| **Levallois** | *0,00* | 2,48 | | 2,41 |
| **unidirectional** | *0,00* | 0,11 | 2,41 | |

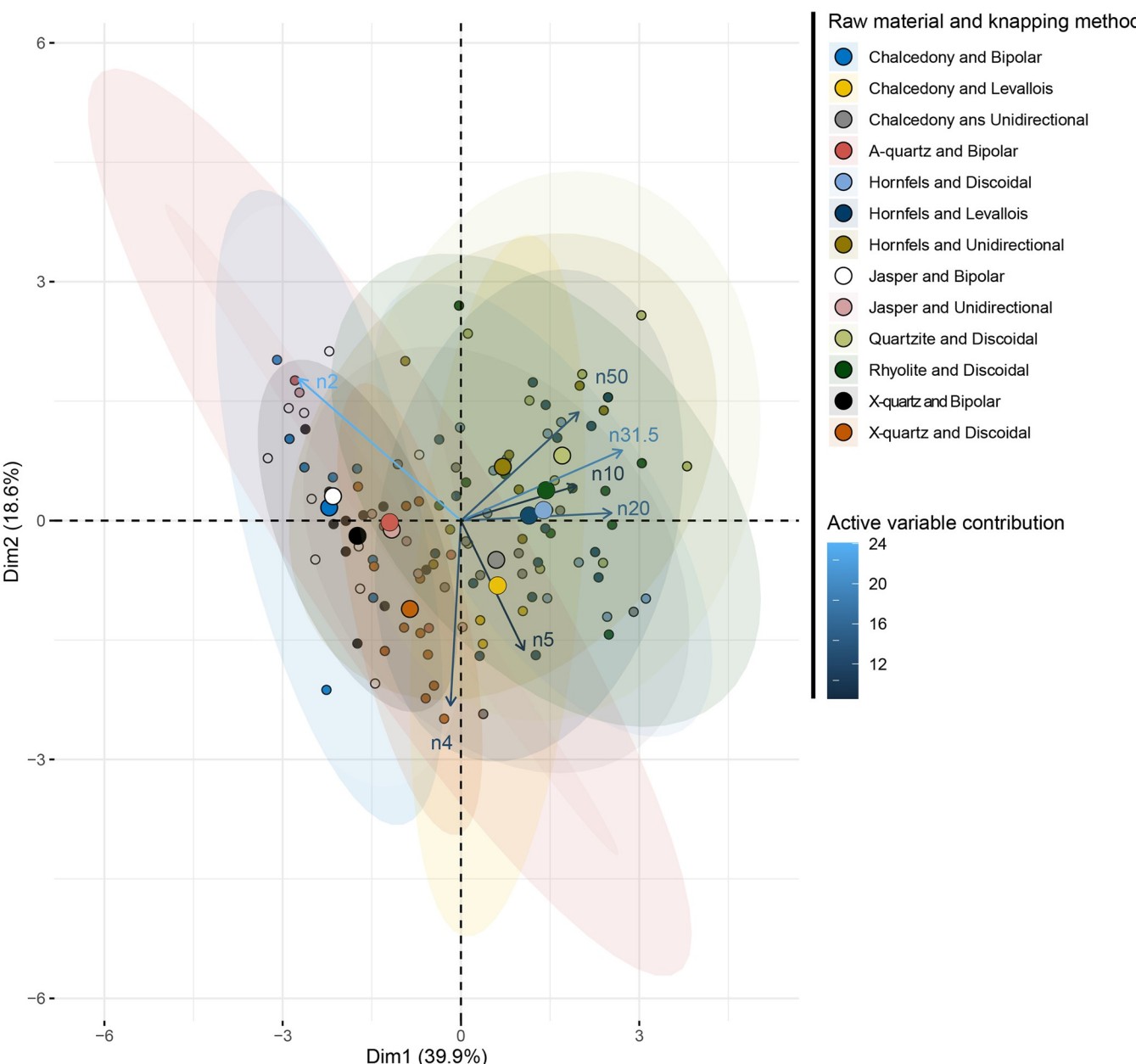

**Fig 12. PCA combining knapping and raw material as a supplementary variable (SA dataset).**

groups appear in the factorial space. The first one is correlated with small particles (i.e., 2 to 4 mm) and is composed of bifacial shaping and laminar knapping systems. A second group, best described as intermediate particle sizes, is composed of bipolar blank production. The third group is best described as having larger particle sizes and is composed of the flaking systems (discoidal, *Levallois* and unidirectional).

When considering raw materials as a supplementary variable (Fig 15 and S2 File), flint is opposed in the factorial space to A-quartz, X-quartz, jasper, and chalcedony, which are all opposed to a group composed of rhyolite, quartzite and hornfels. These three clusters are best described respectively by their small particle sizes (2 to 4 mm), intermediate sizes (4 to 10 mm) and larger sizes (20 to 50 mm).

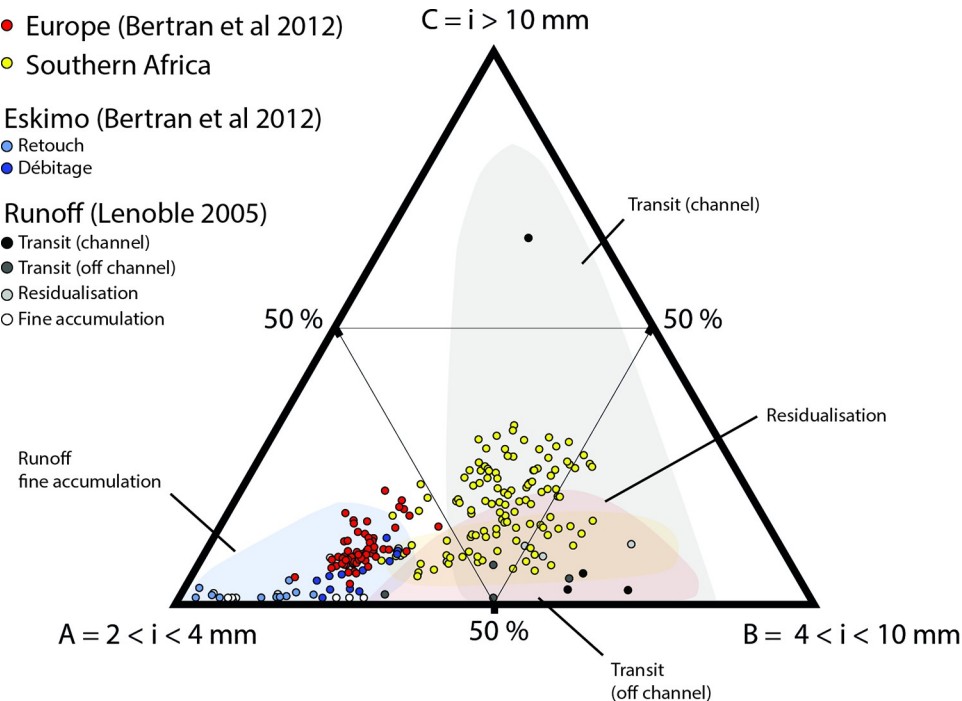

**Fig 13. Ternary plots comparing the SA dataset presented in this paper, the EU and the ESKI datasets published in Bertran et al.** [20, 21] and the RUNOFF dataset [4]. The ternary plots contain 95% confidence intervals.

When adding the ESKI dataset in the principal component analysis with the data divided according to type of reduction, such as "knapping", "shaping" and "retouching" (Fig 16 and S2 File), we see a clear opposition in the factorial space between "retouching", producing smaller remains, and "knapping" producing larger remains. "Shaping" falls between the two.

### Is the particle size distribution of the SA dataset different from the RUNOFF dataset?

The factorial space of the SA dataset compared to the RUNOFF model [4] (Fig 17) shows a clear opposition on the first dimension (53% of the variability) between, on one hand, the fine accumulation area composed of smaller remains (2 to 4 mm size class) and, on the other hand, knapping methods (SA dataset) and runoff areas as residual and transit (in and off channel) described by particles over 4 mm wide. The SA experiments overlap with most of the runoff accumulations of large particles.

## Discussion

### Variability within the SA dataset

Patterson [43]: p. 550 highlighted the need for 'simple analytical techniques' for examining lithic assemblages that can be used by any archaeologist (and not only lithic specialists). This still seems to be a valid point today as lithic assemblages have the advantage of preservation over organic remains, and they provide the most evident continuous cultural record available from the past [44–46].

The results of our experiment highlight two main points within the SA dataset: a distinction between raw materials and a difference between debitage produced by freehand and bipolar knapping techniques.

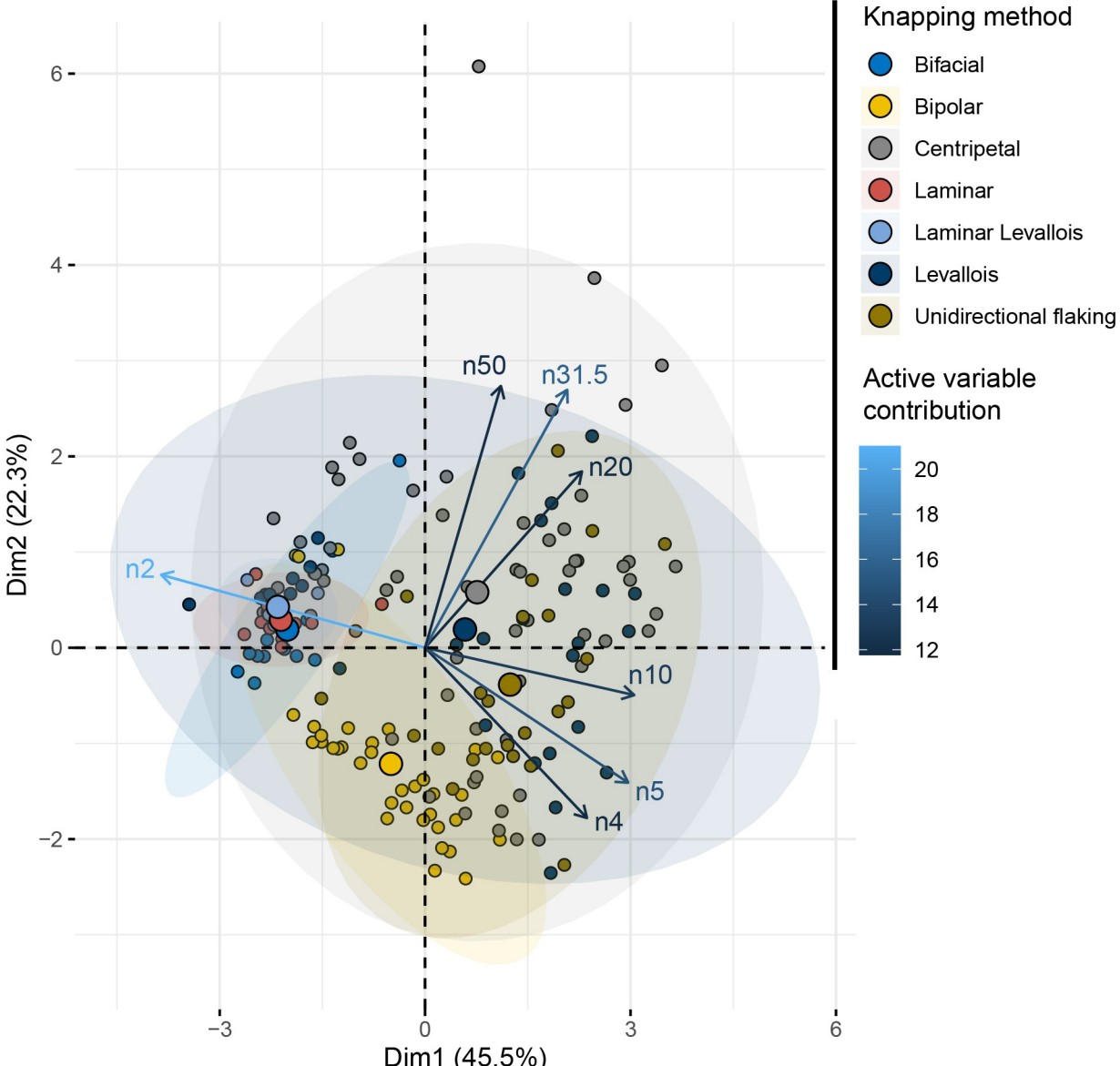

**Fig 14. PCA of the SA dataset compared to the EU dataset [20, 21].** The PCA has been made considering the knapping method as a variable.

Concerning the raw materials, we see, on the one hand, that in our experimental sample chalcedony, X-quartz, A-quartz, and jasper group more or less together and produce smaller size particles; on the other hand, we find rhyolite, quartzite and hornfels producing wider size ranges (Figs 8 and 10). Other authors like K. Schick [2] have also highlighted differences regarding raw materials. Our results prove that experiments using specific raw materials from the regions associated with archaeological assemblages are needed to understand particle size distribution in Stone Age lithic assemblages.

Regarding the distinction between freehand and bipolar knapping methods, from a particle size analysis point of view, it seems possible to distinguish between these two types of reduction. This is in accordance with previous characterizations made to quantitatively distinguish freehand from bipolar axial knapping [13, 47, 48]. This should be tested in future

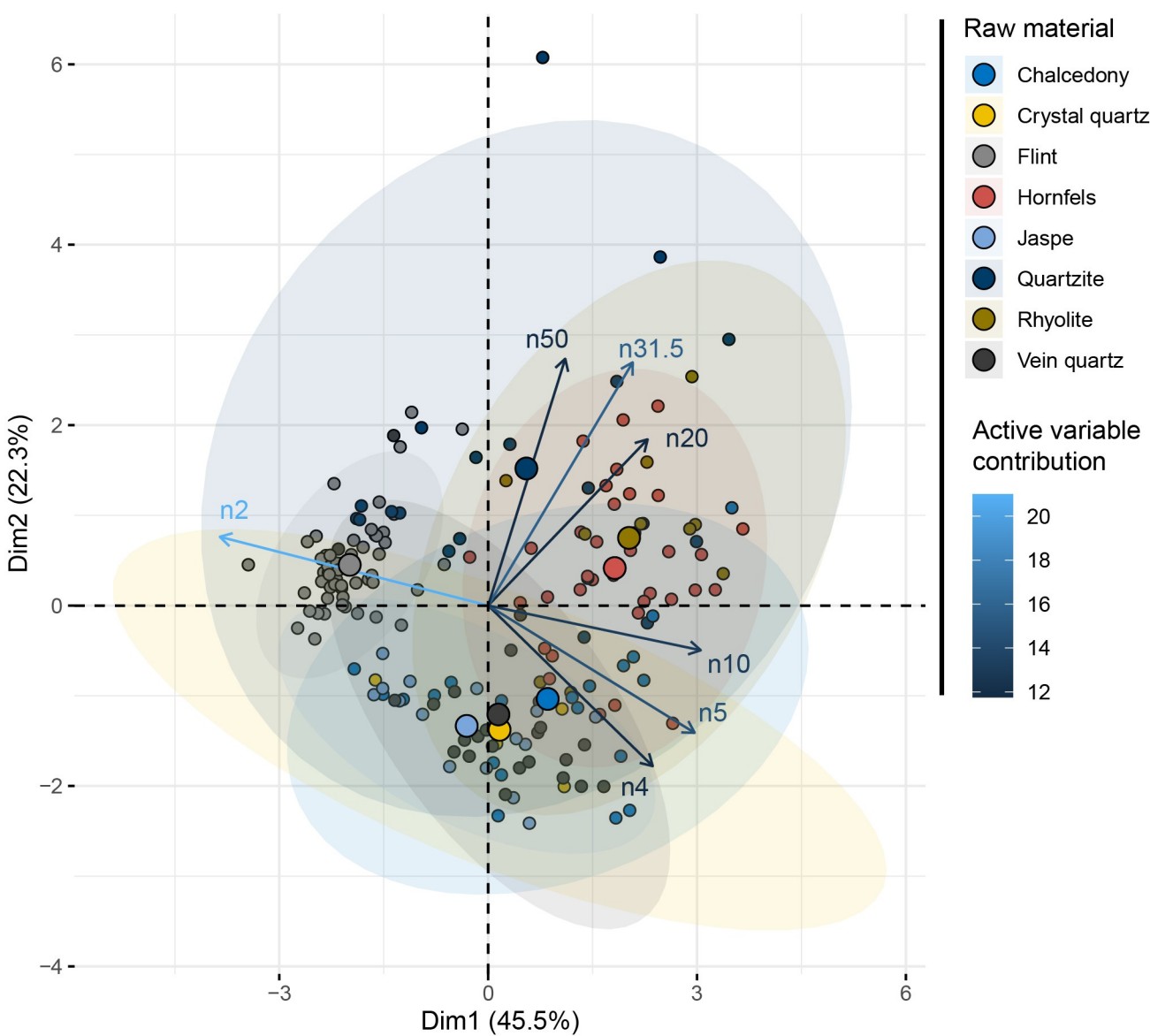

**Fig 15. PCA of the SA dataset and the EU dataset [20, 21], having 'raw material' as a supplementary variable.**

archaeological analyses where freehand and bipolar knapping have been documented within the same assemblage, such as the Howiesons Poort of Sibudu [27, 49], and the miniaturization strategies at Boomplaas [50]. It would be interesting to test if this distinction between freehand and bipolar is feasible considering that the distinction has been informed by the multivariate analysis of the particle size. The reason we mention previous technological analyses is that the main way of identifying bipolar knapping in those studies was through the cores, as it seems the easier way to recognize the reduction technique [51]. If distinguishing between freehand and bipolar knapping by means of particle size is confirmed, this experiment would have made a significant contribution to the field.

Regarding the different freehand knapping methods in our sample, in the factorial space, it does not seem possible to differentiate unidirectional, discoidal and *Levallois* flaking systems. However, as we have not knapped each raw material through all knapping methods, it remains

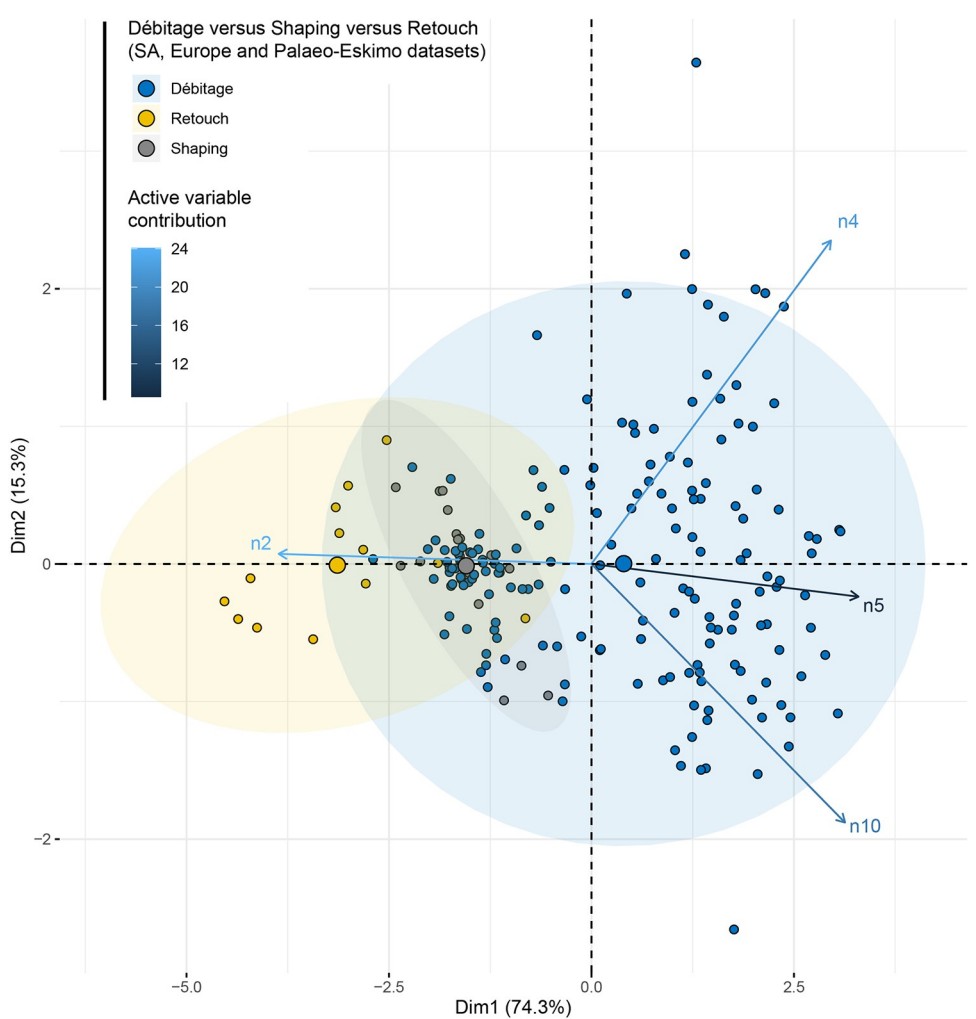

**Fig 16. PCA comparing debitage, shaping and retouch from the SA dataset, EU dataset, and ESKI dataset [20, 21].**

difficult to know what variable (either the knapping method or the raw material) causes these differences. Our results show that the size distribution in southern African archaeological assemblages is knapping method specific and raw material specific.

Due to the variability expressed by the SA dataset, the selection of a single raw material and knapping method appropriate to the site context provides a more accurate model for performing a particle size analysis. Moreover, when combining a specific raw material and a specific knapping method, the distribution areas tend to reduce (with and without 95% confidence intervals) in both ternary plots and principal component analyses.

## Differences between SA, EU and ESKI datasets

Southern African lithic industries produce a greater proportion of bigger particles whatever the raw material and the knapping method (Fig 13). In the southern African dataset, the smallest particles (2 to 4 mm) always constitute 50 to 60% of the reduction sequences. In the EU dataset [20, 21] (see Fig 13 therein) the percentage of this fraction was slightly higher, between 60–70% in all their variants.

It seems that our knapping and raw material particle size distribution differs from the ones published in Bertran and colleagues [20, 21]. In the case of the EU dataset [20, 21], the different

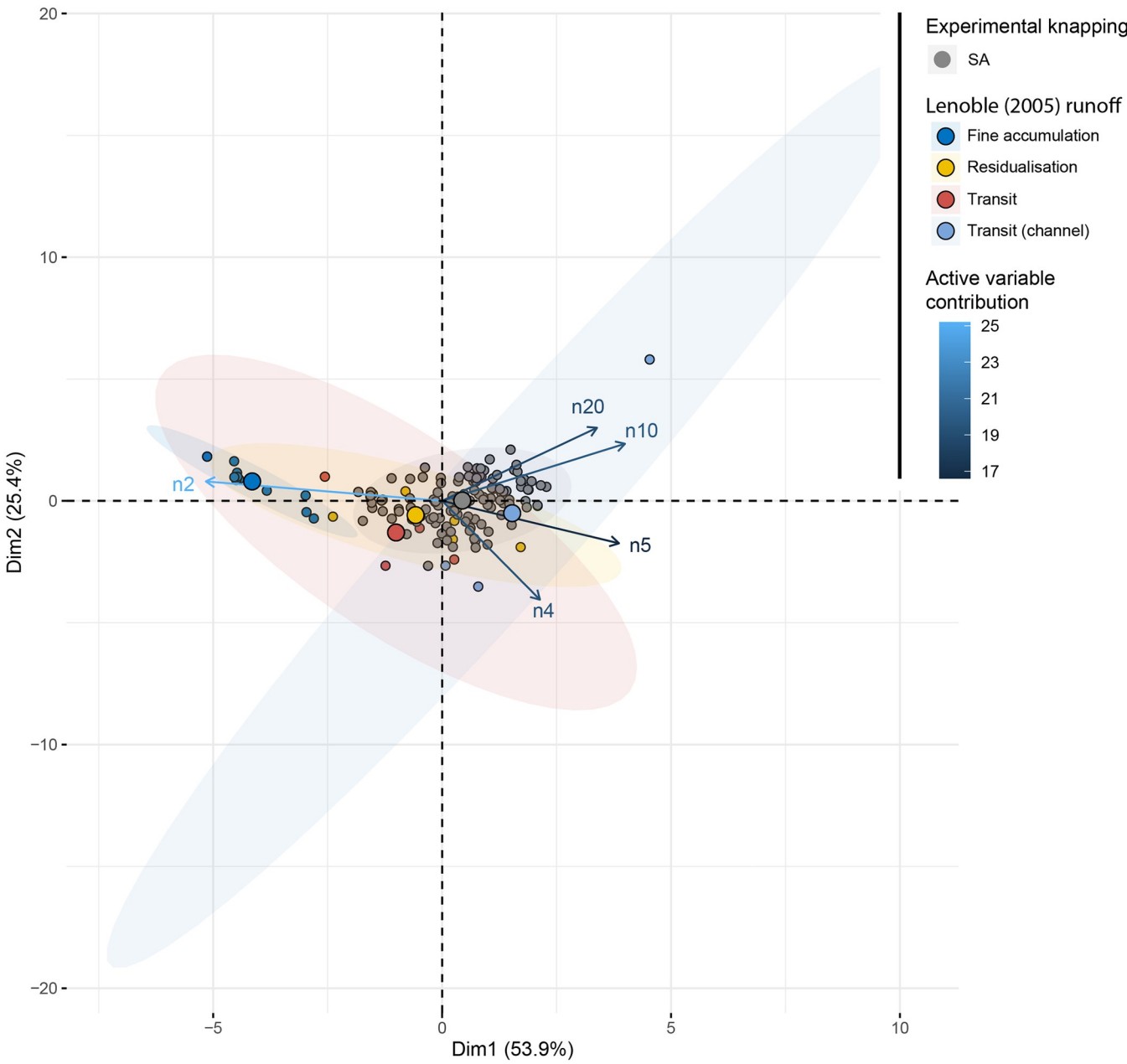

**Fig 17. PCA comparing the SA dataset to the RUNOFF dataset** [4]**.**

experiments overlap remarkably in the multivariate analysis and in the ternary plots (Fig 11), whereas our dataset has a much wider distribution (Fig 13).

The EU, ESKI and SA datasets are clearly separated from each other. The reasons for these different groupings might be due to different aspects. With the evidence at hand, we can only propose some hypotheses that should be tested in future experiments.

Firstly, some technical behaviours produce more small particles. This is particularly the case of microlithic production and retouching activities (ESKI dataset) and to a lesser degree, for bipolar knapping on anvil (SA dataset) and shaping (EU and ESKI datasets). Similar results have been pointed out in other experimental studies [2, 20, 21, 43].

Secondly, the differences between Bertran and colleagues [20, 21] and our sample regarding freehand variants could be the result of the level of skill of the knappers, due to specific knapper gestures and procedures. In other words, it should not exclude the hypothesis that knapper competence and experience have an impact on particle size distribution. Nonetheless, these different levels of knapping performance enrich the models, as it is expected that in the past there would also have been variability in knapping skills within cultural traditions and periods. Moreover, despite similar knapping goals, knapper-specific gestures, habits, and procedures (such as platform preparation or removing striking platform overhang) could also affect size distribution. Again, these variables would require additional experiments.

Thirdly, raw material specificities also induce different patterns in terms of particle size distribution. Whereas flint knapping experiments have a narrow distribution, other types of raw material induce specific distributions with various dispersion features. Regardless of the knapping method, flint tends to produce finer particles than A-quartz, X-quartz, jasper, and chalcedony, which in turn produce finer particles than quartzite, rhyolite and hornfels. Finer raw materials (i.e., flint, jasper, and chalcedony), A-quartz and X-quartz produce finer grain sizes in contrast to coarser metamorphic and volcanic raw materials like quartzite, hornfels and rhyolite.

Finally, considering previous knapping datasets [20, 21] and our knapping experiments, it seems clear that blank transformation ("retouching") produces smaller particles than "shaping" and blank production (debitage). The intensity of retouch and site function [37] also influence the particle size distribution of industries as is already shown in Bertran et al. [20–22]. When a great proportion of the blank is transformed on site, we would expect a greater proportion of the small fraction (2 to 4 mm).

## SA knapping dataset versus RUNOFF experiments

The comparison of the SA dataset distribution and the RUONFF model [4] (for flint) shows a remarkable amount of overlap.

While taken as a whole, the SA dataset shows a broad distribution (contra EU dataset) (Figs 13 and 17). At a site scale, it is nevertheless possible to reduce that distribution by considering only a single knapping method on a single raw material. This will reduce the overlap with the RUNOFF model and allow a more accurate comparison to assess the degree of perturbation in the archaeological record. The overlap between the SA and the RUNOFF dataset could also be due to the different densities of the raw materials. Igneous rocks, such as rhyolite are denser that sedimentary rocks and this could have an impact on this general comparison. This overlapping is also pointing out the need for further experimentation that clarifies the potential reasons for this situation. The new experiments should take into account the geomorphological frequent scenarios for site formation deposits in southern Africa assemblages.

While our experiment has produced very concrete results, the size distribution analysis of lithic material is not sufficient on its own to assess the perturbation of an archaeological lithic assemblage and must be coupled with other analytical tools. In this regard, sedimentary analysis [3], fabric analysis [28, 52, 53], refitting [17, 54–57], spatial analysis [58, 59], material proportions [60], faunal analysis in terms of taphonomy [61], and dating [5] are all tools that can help characterize the impact of natural processes on archaeological assemblages. When all of them are combined [58, 60], we can reach a very detailed site formation model by which to gauge the degree of perturbation of the associated lithic assemblage.

## Conclusion

Our results highlight the need to perform more experimentation to clarify and further support the preliminary distinctions proposed here and the contradictions with previous experimental work.

While all knapping experiments show the same general pattern of size distribution (following a Weibull function, i.e., proportion decreasing as size increases [23]), we have demonstrated that the knapping method, blank transformation activity intensity, and raw material, explain most of the variability. The creation of a new experimental knapping dataset focusing on southern African debitage and raw materials, and its comparison with previous datasets have shown a more complex picture for particle size analysis than was previously thought. There are two main aspects that do not match with recent investigations on particle size analysis in Europe: on the one hand the wider distribution of our experimental knapping, on the other hand the huge overlap between our SA dataset and the RUNOFF dataset. To reduce this overlap, the experimental debitage model compared to the RUNOFF dataset should be knapping method and raw material specific. This means that the performance of a particle size analysis at a site must be done in combination with a technological study of the lithic material. The latter guides the choice of the knapping method and raw material to be included in the comparison model between experimental knapping and runoff experiences on sorting.

## Supporting information

**S1 Data.**
(CSV)

**S1 File. Particle size analyses datasets used in the manuscript.**
(TXT)

**S2 File. Statistical information on the principal component analyses implementations.**
(PDF)

**S3 File. Rmarkdown script with the codes needed to perform the principal component analyses.**
(RMD)

## Acknowledgments

Paloma de la Peña has a *Ramón y Cajal* Research contract (RYC2020-029506-I) at the *Universidad de Granada* (Spain) funded by European social fund and the *Agencia Estatal de Investigación* (Spain).

We thank Gary Trower and Ben Maclennan for indicating to us raw material outcrops and providing us with some of the raw material we used in our knapping experiments. Our thanks also go to Lucinda Backwell, Tammy Hodgskiss, Matt Caruana and Matt Lotter, whose reading and correction of the article were very valuable.

## Author Contributions

**Conceptualization:** Paloma de la Peña, Marc Thomas.

**Data curation:** Paloma de la Peña, Tumelo R. Molefyane.

**Formal analysis:** Paloma de la Peña, Marc Thomas, Tumelo R. Molefyane.

**Investigation:** Paloma de la Peña, Marc Thomas, Tumelo R. Molefyane.

**Methodology:** Paloma de la Peña, Marc Thomas, Tumelo R. Molefyane.

**Project administration:** Paloma de la Peña, Marc Thomas.

**Resources:** Paloma de la Peña.

**Software:** Marc Thomas.

**Supervision:** Paloma de la Peña, Marc Thomas.

**Writing – original draft:** Paloma de la Peña, Marc Thomas.

**Writing – review & editing:** Paloma de la Peña, Marc Thomas.

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
