## [Decision Letter · Decision Letter 0]

23 May 2022

PONE-D-22-10277Particle size distribution of Southern African rocks and methods of debitagePLOS ONE

Dear Dr. De la Pena,

Thank you for submitting your manuscript to PLOS ONE. After careful consideration, we feel that it has merit but does not fully meet PLOS ONE’s publication criteria as it currently stands. Therefore, we invite you to submit a fully revised version of the manuscript that precisely addresses the points raised during the review process.

We look forward to receiving your revised manuscript.

Kind regards,

Enza Elena Spinapolice, Ph.D

Academic Editor

PLOS ONE

Journal Requirements:

"PdlP received CoE-NRF (South Africa) funding"

"This work has been supported by the Center of Excellence of Paleosciences - National Research Foundation of South Africa"

"PdlP received CoE-NRF (South Africa) funding"

6. Please upload a copy of supplementary material 1 (S1) which you refer to in your text.

Reviewers' comments:

Reviewer's Responses to Questions

**Comments to the Author**

1. Is the manuscript technically sound, and do the data support the conclusions?

Reviewer #1: Partly

Reviewer #2: Partly

2. Has the statistical analysis been performed appropriately and rigorously? 

Reviewer #1: Yes

Reviewer #2: Yes

3. Have the authors made all data underlying the findings in their manuscript fully available?

Reviewer #1: Yes

Reviewer #2: No

4. Is the manuscript presented in an intelligible fashion and written in standard English?

Reviewer #1: No

Reviewer #2: Yes

5. Review Comments to the Author

Reviewer #1: This paper by de la Pena et al. is a welcome addition to the literature on experimental studies on particle size distribution. The major contribution of the article is the presentation of an open-access database on rocks used frequently in the MSA and LSA of southern Africa for size distribution from an experimental knapping program. The study is overall well-done and scientifically sound – following clear research questions and a clear presentation of results - and I think an open-access journal like PLOS is the perfect place for such a study. I do, however, have a couple of critical remarks that I would urge the authors to consider and incorporate into the manuscript before it can be published.

My biggest criticism concerns the methodical setup of the experimental knapping. There is little explanation or context for what the initial set-up was: Why were 117 cores knapped (how to get to that number)? How was the distribution across raw materials and reduction methods arrived at? Was there any bigger experimental design behind it? Looking at the distribution of cores in Table 1, it honestly looks a bit unsystematic. The reasoning that not all knapping methods were not applied to all raw materials due to the Southern African archaeological context (lines 210-212) is not convincing, as there is evidence for all of the knapping methods on all raw materials of this study (admittedly much rarer for some raw materials, e.g. Levallois on quartz is rare). This then also compromises some statistical analyses of the study as the authors themselves acknowledge (lines 213-214). Ideally, the study would incorporate some more reduction of different raw materials/knapping methods or be at the very list much clearer about the description of the study design in the Materials and Methods section!

While the results are well-done and well-presented, the discussion is somewhat lacking substance and not yet sufficient for publication. Most importantly, the authors should at least provide one test application (i.e. case study) showing the value of their experimental study and the particle size distribution in one MSA or LSA assemblage from southern Africa. This would clearly show the analytical value of the current study for archaeology, which is so far only assumed but not shown. Particularly, this could show with empirical data why their database and not the EU dataset should be used for such an analysis. Additional discussion on the potential insights that can be gained from the experimental database for MSA/LSA studies from a general potential would also much add to the relevance of the current study.

Finally, some parts of the manuscript such as the Introduction (lines 48-54) and several parts later om need some further polishing of the English language, potentially by the additional help of a native speaker.

Reviewer #2: The paper submitted by Dr. de la Pena and colleagues entitled "Particle size distribution of Southern African rocks and methods of débitage" is a fair contribution to the question regarding lithic assemblages distribution within archaeological contexts. Especially because it focuses on raw material selection and knapping methods from Southern Africa contexts.

However, it needs to be revised since it is necessary to provide more information about the methods applied and the authors' choice that push them to perform this research.

The paper would benefit of a more accurate description of the reason why a South African dataset has been compared with European datasets.

The authors assess that particle size distribution variability is strongly affected by raw materials and knapping methods, however it is not clearly explained if to demonstrate this diversity was the main reason behind their choice to perform comparative analyses between different contexts and different raw materials. I think that much benefit will be provided by clarifying since the beginning of the paper what is the main purpose of this research.

It is true that raw material variability strongly affects the results of these type of analysis, thus my question is if it is worth it to compare different débitage methods performed on different raw materials from different geographical contexts. It would be useful if the authors will dedicate a section in the paper to explain this main decision.

Furthermore, the statistical analyses are performed in the correct way. However, it is quite difficult to navigate through the PCA results without knowing any information regarding the loadings and the scores that each principal component provides. In this case, the graphs alone it is not enough to understand the degree of variance of the analyzed variables.

I would appreciate an additional table with variables used for comparative analyses, including the variables used from the datasets from previous experimental analyses. At this point, there are no information regarding the normalization of the data between the different datasets.

I recommend major revision taking into account the advice to make clearer the description of the research and the reasons behind it.

Additional comments are provided within the main manuscipt.

6. PLOS authors have the option to publish the peer review history of their article (what does this mean?). If published, this will include your full peer review and any attached files.

Reviewer #1: No

Reviewer #2: No

---

## [Author Response · Author response to Decision Letter 0]

5 Sep 2022

Dear Editor,

Thank you for letting us revise the manuscript. We are also grateful for the constructive comments of the two reviewers. We have addressed all the comments and problems highlighted in the review.

Yours sincerely,

Dr. Paloma de la Peña

In Johannesburg, 11th August 2022

Response to reviewers

Reviewer #1

Have the authors made all data underlying the findings in their manuscript fully available?

We incorporated our database into the supplementary material (SM1) and added some statistical information on the implementation of PCA (SM2) and an R script that allows us to reproduce all PCA analyses and obtain more statistics (SM3).

Is the manuscript presented in an intelligible fashion and written in standard English?

The manuscript was revised in the first and second submission by Dr Tammy Hodgskiss and in the second revision by Dr Lucinda Backwell, both of them are native English-speaking persons very familiar with archaeology vocabulary as they regularly publishes on Middle and Later Stone Age Archaeology. 

My biggest criticism concerns the methodical setup of the experimental knapping. There is little explanation or context for what the initial set-up was: Why were 117 cores knapped (how to get to that number)? How was the distribution across raw materials and reduction methods arrived at? Was there any bigger experimental design behind it? Looking at the distribution of cores in Table 1, it honestly looks a bit unsystematic. The reasoning that not all knapping methods were not applied to all raw materials due to the Southern African archaeological context (lines 210-212) is not convincing, as there is evidence for all of the knapping methods on all raw materials of this study (admittedly much rarer for some raw materials, e.g. Levallois on quartz is rare). This then also compromises some statistical analyses of the study as the authors themselves acknowledge (lines 213-214). Ideally, the study would incorporate some more reduction of different raw materials/knapping methods or be at the very list much clearer about the description of the study design in the Materials and Methods section!

Thank you for this comment. The reviewer is right that the experiment seems a bit unsystematic. The number of experiments is odd because if the core fragmented during the knapping process, we did not include those experiments. Moreover, it has also been influenced by the covid pandemic. The experiment was done in late 2020 and early 2021 when it was not possible to access raw material outcrops due to driving and transport restrictions, so we operate with the material available at the laboratory of Archaeology at the Evolutionary Studies Institute. 

Finally, the reason for some raw materials and knapping varieties not represented is that there are some knapping methods that are not reported in the archaeological record (Middle and Later Stone Age). For instance, to the best of our knowledge there is no evidence of Levallois recurrent centripetal reduction on automorphic and xenomorphic quartz or, on the other hand, there is no documentation on bipolar knapping on hornfels. It would have no archaeological sense to include those varieties in the experimental collection.

This has been clarified in the presentation of the methodology experimentation.

While the results are well-done and well-presented, the discussion is somewhat lacking substance and not yet sufficient for publication. Most importantly, the authors should at least provide one test application (i.e. case study) showing the value of their experimental study and the particle size distribution in one MSA or LSA assemblage from southern Africa. This would clearly show the analytical value of the current study for archaeology, which is so far only assumed but not shown. Particularly, this could show with empirical data why their database and not the EU dataset should be used for such an analysis. Additional discussion on the potential insights that can be gained from the experimental database for MSA/LSA studies from a general potential would also much add to the relevance of the current study.

We have reinforced and re-done the discussion adding new ideas and an in depth criticism of our results (see the changes in the final section)

We think it is a very good idea to compare our experimental results with archaeological collections and, indeed, we would like to do another research paper only for that comparison. We wanted first to publish the experimental material and to compare it to other experimental studies such as Schick (1986, 1987) or Bertran et al. (2006, 2012), which are valuable contributions for African and European assemblages respectively. 

Finally, some parts of the manuscript such as the Introduction (lines 48-54) and several parts later om need some further polishing of the English language, potentially by the additional help of a native speaker.

The manuscript was revised in the first and second submission by Dr Tammy Hodgskiss and in the second revision also by Dr Lucinda Backwell, both of them are native English speaking persons very familiar with archaeology vocabulary as they regularly publishes on Middle and Later Stone Age Archaeology. 

Reviewer #2

The paper would benefit of a more accurate description of the reason why a South African dataset has been compared with European datasets.

The authors assess that particle size distribution variability is strongly affected by raw materials and knapping methods, however it is not clearly explained if to demonstrate this diversity was the main reason behind their choice to perform comparative analyses between different contexts and different raw materials. I think that much benefit will be provided by clarifying since the beginning of the paper what is the main purpose of this research.

It is true that raw material variability strongly affects the results of these type of analysis, thus my question is if it is worth it to compare different débitage methods performed on different raw materials from different geographical contexts. It would be useful if the authors will dedicate a section in the paper to explain this main decision.

The only available datasets made thanks to the Bertran et al methodology describe mainly European archaeological context. That was the main reason of this comparison and doing these experiments and the datasets on particle size. Moreover, particle size analysis in southern Africa archaeological collections have not been applied before. To have a valid framework it needs to be addressed from the most frequent raw materials and the knapping methods in this chronology.

The main purpose of the paper is to offer a framework to understand the particle size distribution on Southern African sites of the Late Pleistocene. Following this reviewer advice we have reinforced this in the manuscript (see the abstract, the introduction and the “Questions and objectives” sections.

Furthermore, the statistical analyses are performed in the correct way. However, it is quite difficult to navigate through the PCA results without knowing any information regarding the loadings and the scores that each principal component provides. In this case, the graphs alone it is not enough to understand the degree of variance of the analyzed variables.

I would appreciate an additional table with variables used for comparative analyses, including the variables used from the datasets from previous experimental analyses. At this point, there are no information regarding the normalization of the data between the different datasets

We have included the statistics information associated to the PCA in SM 2 and 3.

We have also included in SM1 the datasets of previous experimental analyses together with our own experimentation, everything studies and formatted in the same way (sieving with the same screen mesh for at least 4 size class).

...

We have also modified figures 1, 6 and 14 as there were some typos. 

Finally, we would like to change our title to: Particle size distribution: an experimental study using southern African reduction methods and raw materials

---

## [Decision Letter · Decision Letter 1]

24 Oct 2022

PONE-D-22-10277R1Particle size distribution: an experimental study using southern African reduction methods and raw materialsPLOS ONE

Dear Dr. De la Pena,

Thank you for submitting your manuscript to PLOS ONE. After careful consideration, we feel that it has merit but does not fully meet PLOS ONE’s publication criteria as it currently stands. Therefore, we invite you to submit a revised version of the manuscript that addresses the points raised during the review process.

We look forward to receiving your revised manuscript.

Kind regards,

Enza Elena Spinapolice, Ph.D

Academic Editor

PLOS ONE

Journal Requirements:

Reviewers' comments:

Reviewer's Responses to Questions

**Comments to the Author**

1. If the authors have adequately addressed your comments raised in a previous round of review and you feel that this manuscript is now acceptable for publication, you may indicate that here to bypass the “Comments to the Author” section, enter your conflict of interest statement in the “Confidential to Editor” section, and submit your "Accept" recommendation.

Reviewer #1: All comments have been addressed

Reviewer #2: All comments have been addressed

2. Is the manuscript technically sound, and do the data support the conclusions?

Reviewer #1: Yes

Reviewer #2: Yes

3. Has the statistical analysis been performed appropriately and rigorously? 

Reviewer #1: Yes

Reviewer #2: Yes

4. Have the authors made all data underlying the findings in their manuscript fully available?

Reviewer #1: Yes

Reviewer #2: Yes

5. Is the manuscript presented in an intelligible fashion and written in standard English?

Reviewer #1: Yes

Reviewer #2: Yes

6. Review Comments to the Author

Reviewer #1: The authors have modified the manuscript accordingly to the comments by myself and/or responded to the criticisms in a sufficient manner. I support acceptance of the article in its current form.

Reviewer #2: The paper by de la Pena and colleagues, which aims to propose a particle size analysis with a focus on southern Africa MSA and LSA, has significantly improved. The objectives and aims of this research are better defined thus the results and conclusions have the benefit of a stronger introduction.

I appreciated the supplementary materials provided, as well as the R codes to replicate statistical analysis. It is way much easier to navigate through the multivariate analyses now.

However, I think that the abstract could be even improved. This section doesn’t fully explain the potentiality of this research, in particular, I would suggest revising it by adding at the end of the section a couple of sentences that explain the importance of this study. I added a comment within the pdf file.

I added a few more comments and suggestions within the Pdf file. There are some sentences that need to be revised since they generate confusion during the reading, and I suggest revising the text before publication.

I recommend publication with minor revisions.

7. PLOS authors have the option to publish the peer review history of their article (what does this mean?). If published, this will include your full peer review and any attached files.

Reviewer #1: **Yes: **Manuel Will

Reviewer #2: No

---

## [Author Response · Author response to Decision Letter 1]

20 Nov 2022

Dear Editor,

Thank you for letting us revise the manuscript. We are also grateful for the constructive

comments of second reviewer. We have addressed all the comments and problems

highlighted in the pdf. In the pdf provided by the reviewer we have answered all her/his comments (see attachment 1 for editor)

Yours sincerely,

Dr. Paloma de la Peña

---

## [Editor Report · Decision Letter 2]

28 Nov 2022

Particle size distribution: an experimental study using southern African reduction methods and raw materials

PONE-D-22-10277R2

Dear Dr. De la Peña,

We’re pleased to inform you that your manuscript has been judged scientifically suitable for publication and will be formally accepted for publication once it meets all outstanding technical requirements.

Kind regards,

Enza Elena Spinapolice, Ph.D

Academic Editor

PLOS ONE

---

## [Editor Report · Acceptance letter]

6 Dec 2022

PONE-D-22-10277R2 

Particle size distribution: an experimental study using southern African reduction methods and raw materials 

Dear Dr. de la Peña:

I'm pleased to inform you that your manuscript has been deemed suitable for publication in PLOS ONE. Congratulations! Your manuscript is now with our production department. 

Kind regards, 

on behalf of

Dr. Enza Elena Spinapolice 

Academic Editor

PLOS ONE